# The Value of Ultrasound in the Evaluation of the Integrity of Silicone Breast Implants

**DOI:** 10.3390/medicina57050440

**Published:** 2021-05-03

**Authors:** Dalia Rukanskienė, Greta Bytautaitė, Agnė Česnauskaitė, Loreta Pilipaitytė, Tautrimas Aštrauskas, Eglė Jonaitienė

**Affiliations:** 1Clinic of Radiology, Hospital of Lithuanian University of Health Sciences, LT-50161 Kaunas, Lithuania; drebule@hotmail.com; 2Tautrimas Aštrauskas Clinic, LT-45325 Kaunas, Lithuania; loreta.pilipaityte@gmail.com (L.P.); info@tautrimas.com (T.A.); 3Faculty of Medicine, Medical Academy, Lithuanian University of Health Sciences, LT-44307 Kaunas, Lithuania; gretabytaut@gmail.com (G.B.); a.cesnauskaite@gmail.com (A.Č.); 4Department of Plastic and Reconstructive Surgery, Medical Academy, Lithuanian University of Health Sciences, LT-44307 Kaunas, Lithuania; 5Clinic of Plastic and Reconstructive Surgery, Hospital of Lithuanian University of Health Sciences, LT-50161 Kaunas, Lithuania; 6Department of Radiology, Medical Academy, Lithuanian University of Health Sciences, LT-44307 Kaunas, Lithuania

**Keywords:** silicone breast implants, ruptured implant, intact implant, ultrasound, magnetic resonance imaging

## Abstract

*Background and Objective:* Breast implant surgery for cosmetic purposes is the most popular plastic surgery and it has been performed for over 100 years. Rupture of silicone gel-filled breast implants usually is asymptomatic and is one of the more dangerous complications due to free silicone migration. The aim of our study was to evaluate the diagnostic value of ultrasound (US) in the evaluation of the integrity of silicone breast implants and identify the main sign of intact and ruptured breast implants. *Patients and Methods:* In this retrospective study, the medical documentation of women who underwent breast implant surgery and US checkups at Tautrimas Aštrauskas Clinic in Kaunas, Lithuania, during 2015–2020 was analyzed. The patients were divided into two groups: patients with intact and ruptured breast implants. The accuracy, sensitivity, specificity, positive predictive value (PPV), and negative predictive value (NPV) as well as the signs of implant integrity of US examination were evaluated. *Results:* In this study, 76 women with bilateral breast implants (*n* = 152) were reviewed. On a US examination, ruptured implants were found in 41.1% (*n* = 61) of the cases; of them, 78.7% (*n* = 48) of the cases had ≥2 US signs of a ruptured implant, and in all these cases, implant rupture was confirmed at surgery. Overall, one US sign of a ruptured implant was found in 21.3% (*n* = 13) of the cases. Of them, inhomogeneous content in all cases (*n* = 3) was found in the intact implant group, and an abnormal implant shell was documented more often in the ruptured implant group, not intact one (*n* = 9, 90% vs. *n* = 1, 10%). US had a diagnostic accuracy of 94.7%, sensitivity of 98.3%, specificity of 89.2%, PPV of 93.4%, and NPV of 97.1% in the evaluation of implant integrity. *Conclusions:* Our results show that US is a very reliable alternative in evaluating breast implant integrity and could be the investigation of choice for implant rupture, while MRI could be advocated only in inconclusive cases. Uneven implant shell was found to be the most important US sign of breast implant rupture. Based on the findings, we recommend performing US examination after breast augmentation surgery with silicone gel-filled implants annually.

## 1. Introduction

Breast implant surgery for cosmetic enlargement or reconstruction is the most popular plastic surgery and has a history of over 100 years. The records of the American Society of Plastic Surgeons show a constantly increasing number of breast implant surgeries, which particularly has risen during the past decade, and in 2019 there were 299,715 breast augmentation surgeries in the United States (approximately 1 for 547 women) [1].

The most common local complications occurring after breast implant surgery using silicone gel-filled implants are capsular contracture and rupture of the implant [2]. The rate of implant ruptures increases with time, and most of them do not cause any clinical symptoms [3,4]. Once an implant ruptures, free silicone can migrate. Most frequently free silicone infiltrates the adjacent breast tissues and sometimes can mimic breast cancer [5,6]. The second most common place for free silicone migration is regional lymph nodes (axillary lymph nodes), and silicone aggregates in lymph nodes can also mimic malignant processes [7,8]. Occasionally, free silicone travels to distant regions (arm/forearm, thoracic cavity, abdominal wall, legs, back) [9,10]. In order to avoid these complications, it is of crucial importance to detect implant rupture as soon as possible and to remove or replace a ruptured implant [2].

In the assessment of appropriate breast implant evaluation by the American College of Radiology, magnetic resonance imaging (MRI) is considered the “gold” standard for breast implant screening [11]. However, due to lower availability and considerably higher costs of MRI, most surgeons do not follow the FDA recommendations and explore the possibilities to use other imaging modalities for routine screening of asymptomatic women [12,13,14]. Moreover, the new guidelines from the U.S. Food and Drug administration consider breast ultrasound (US) of an equal value for the detection of asymptomatic breast implant rupture [15]. Numerous studies have compared the diagnostic value of US and MRI in the evaluation of breast implant integrity [3,14,16,17,18,19,20]. However, the sensitivity of US reported by different authors varies considerably, i.e., from 50% [17] up to 94% [20]. Therefore, one of the aims of our study was to determine if US is reliable in the evaluation of breast implant integrity. Based on the recommendations by Hold et al., Bogetti et al., and Zingaretti et al. [17,19,21], if US examinations showed definite implant rupture, no MRI was performed, and if US findings were inconclusive, only then women were subjected to MRI.

There are several literature sources describing the signs of intact and ruptured implants on US. An intact implant on US shows an anechoic interior and a smooth contour. The signs of intracapsular rupture on US are as follows: (1) the “keyhole” or “noose” sign, (2) the subcapsular line sign, (3) the “stepladder” sign, and (4) inhomogeneous implant content. Silicone granulomas outside an implant shell (breast tissues, axillary lymph nodes) that may appear as anechoic cysts, isoechoic solid nodules, or classic snowstorm artifacts on US are the sign of extracapsular implant rupture [22,23,24]. If two or more signs of implant rupture are observed on US, there should be no doubts that the implant is ruptured. More questions about implant integrity arise when only one US sign of a ruptured implant is found; therefore, another aim of this study was to identify the most important sign of a ruptured implant.

## 2. Materials and Methods

A retrospective study of medical documentation of women who had US examinations and underwent silicone breast implant surgery at Tautrimas Aštrauskas Clinic in Kaunas, Lithuania, from 2015 to 2020 was approved by the Kaunas Regional Biomedical Research Ethics Committee (permission No. BE-10-3, approved on 3 March 2020).

US examinations were performed with a SonoScape S6/S6Pro/S6BW US unit using a 12-MHz linear transducer at two-dimensional (B2) mode. MRI was performed using an MRI scanner (Siemens MAGNETOM^®^ Avanto 1.5 T) with a dedicated breast coil. The following MRI sequences for the examination of implant morphology and integrity were acquired: (1) the high-resolution T2-weighted axial sequences, (2) water-suppressed fast spin-echo T2-weighted axial sequences, and (3) the axial inversion recovery (IR) sequence with water saturation (“silicone only”) sequences. Sagittal sequences were acquired on the side of the implant being investigated. No oral or intravenous contrast was given. US and MRI examinations assessed whether breast implants were intact or ruptured and whether the axillary lymph nodes were normal or silicone containing.

All US examinations were performed and MRI examinations were evaluated by one radiologist with 13-year experience in breast diagnostics and 5-year experience in diagnostics of the augmented breast.

The inclusion criteria were as follows: (1) breast correction surgery with silicone implants, (2) breast US examination, (3) intact implant diagnosed by US, and (4) implant rupture by US was followed by implant replacement surgery. The exclusion criteria were (1) implant rupture detected/suspected by US, but no implant replacement surgery was performed, (2) the contracture of the implant, and (3) breast US was not performed.

US examination signs of an intact and a ruptured breast implant were chosen with reference to literature and examined breast implants were divided into two groups—intact and ruptured [23,24]. An intact breast implant was characterized by an even and continuous implant shell and homogeneous intracapsular echotexture (Figure 1).

The signs of a ruptured breast implant were an uneven implant shell and inhomogeneous intracapsular echotexture (Figure 2).

Axillary lymph nodes were assessed as normal (oval shape, smooth cortex, unchanged, clearly visible fat gate) or containing silicone (a sharp upper border and sides, a loss of the lower border due to the “snowstorm” artifact) (Figure 3).

A total of 152 breasts with implants were included in the study (76 women with bilateral breast implants). As many as 28 patients with 56 implants had intact breast implants, which were confirmed by two US checkups within 2–3 years, whereas 48 patients (96 implants) had either one or both ruptured implants. If rupture on US was questionable, a breast MRI was performed. Once implant rupture was detected by US and/or MRI, it was followed by surgery. If intraoperatively the implant shell was undamaged and there was no free silicone outside the implant shell, the implant was defined as intact, and if there was a shell defect and silicone was present outside the implant shell, the implant was considered as ruptured.

If one implant was found to be ruptured, the second one was also replaced in the following cases: (1) initial implant surgery was performed more than 10 years ago, (2) if that type of implant was no longer on the market, and (3) if the patient requested a replacement. In other cases, only the damaged implant was replaced.

Statistical analysis was performed using IBM SPSS Statistics 25 package (IBM Corp. Released in 2017, IBM SPSS Statistics for Windows, Version 25.0, Armonk, NY, USA: IBM Corp). Descriptive statistics methods were used to systemize the study data. Nonparametric Mann–Whitney test was used for the comparison of two independent groups of continuous data that do not follow a normal distribution. The chi-square test for independence (homogeneity) was used for the analysis of nominal characteristics, and Fisher’s exact test was applied when the frequency in at least one cell was small. The results are described by the frequency of the qualitative attribute values and the relative frequency in the comparative samples. The accuracy, sensitivity, specificity, positive predictive value (PPV), and negative predictive value (NPV) of the US examination were evaluated, and the estimates of these characteristics were given, along with their 95% confidence intervals (CIs). Odds ratio and 95% CIs were used to evaluate the risk. The observed differences and associations were considered statistically significant if *p* < 0.05.

## 3. Results

A total of 76 women with 152 silicone breast implants were reviewed in this study (Figure 4). The age of the patients ranged between 23 and 60 years; the mean age was 37 years (SD, 0.61). On a US examination, there were 91 intact implants (59.9%), and 61 implants (41.1%) were found or were suspected to be ruptured. Implant replacement surgery was performed in 62.5% (*n* = 95) of the cases (37 implants were intact and 58 were ruptured), and in the remaining 37.5% (*n* = 57) of the cases, no surgery was performed. Intact implants were found in 61.8% of cases (*n* = 94), and 58 implants were ruptured (38.2%). The majority (79.3%, *n* = 46) of implant ruptures were intracapsular, and the rest (20.7%, *n* = 12), extracapsular (with axillary lymph nodes containing silicone). Normal lymph nodes were found in 90.8% (*n* = 138), and lymph nodes with silicone-containing features were found in 9.2% (*n* = 14) of all cases.

Breast MRI was performed in 17.1% of cases (*n* = 26). Of them, 14 (53.8%) were intact based on the findings on MRI and confirmed during surgery, while in 46.2% (*n* = 12), MRI showed a ruptured implant (Figure 4). However, the rupture was confirmed during surgery in 11 cases. One flipped-over but not ruptured implant was detected (Figure 5).

The median time that has passed since primary breast implant surgery for intact and ruptured implants is shown in Table 1.

The median time since the primary implantation surgery for intact and ruptured implants was 4 and 8 years, respectively (*p* < 0.001). The likelihood of implant rupture increased by 1.101 times each year after breast implant surgery.

Overall, the accuracy and NPV of US for implant rupture were 96.7% and 98.9%, respectively, and only 0.7% of the cases were false negative. In the case of implant ruptures confirmed by surgery, the accuracy and NPV of US were lower, i.e., 94.7% and 97.1%, respectively, and false-negative results were found in 1.1% of these cases (Table 2).

Our study established that the shell and content of the implant as well as findings in axillary lymph nodes differ for intact and ruptured implants (*p* < 0.001) (Table 3).

The shell of ruptured implants was more frequently abnormal than that of intact implants (98.3% vs. 1.1%), the content similarly was more often inhomogeneous in ruptured than undamaged implants (82.8% vs. 3.2%), and silicone-containing lymph nodes were detected more often in ruptured than intact implants (20.7% vs. 2.1%).

Table 4 shows the distribution of intact and ruptured implants by the number of signs of an intact and a ruptured implant on US examination.

In the case of an intact implant, all three US signs of an intact implant—even implant shell, homogeneous content, and normal axillary lymph nodes—were observed most frequently (93.6%). In the remaining cases, implants had one US sign of a ruptured implant: one implant had an abnormal implant shell, three implants showed inhomogeneous content, and two implants appeared as undamaged, but axillary lymph nodes contained silicone. In the group of ruptured implants, only in one case (1.7%), there was no single sign of rupture on US. ≥2 signs of implant rupture on US were observed in 48 cases (82.8%), and only one sign of implant rupture on US was documented in 9 cases (15.5%), and it was an abnormal implant shell (Figure 6).

Analysis of these results showed that during US examination, only one sign of implant rupture was found in 15 cases: of them, 2 cases had silicone in lymph nodes (the implant itself looked intact), and the remaining 13 cases had only one sign of implant rupture. Of them, only inhomogeneous content was found in 3 cases (the implant was intact in all these cases), and in 10 cases, US showed only abnormal implant shell (on surgery, 9 cases had a ruptured implant, and in 1 case, the implant was not ruptured just flipped over (*n* = 9, 90% vs. *n* = 1, 10%)).

## 4. Discussion

This study had a dual aim to (1) determine the diagnostic value of US in the evaluation of implant integrity and (2) identify which US signs of an intact and a ruptured implant are most important. Our study showed that US is very accurate in the evaluation of implant integrity (diagnostic accuracy of 94.7%, sensitivity of 98.3%, NPV of 97.1%), and it has to be a first-choice imaging modality. We recommend performing US every year after breast augmentation surgery since an implant can rupture earlier than expected. If ≥2 signs of a ruptured implant are detected on US, we can rely on US examination by 100%. If only one sign of a ruptured implant is found, it is advocated to perform MRI as well. The most important sign of a ruptured implant on US is an abnormal implant shell as the implant content can become inhomogeneous without any implant damage.

In the majority of studies, the performance of two imaging techniques—US and MRI—is compared [3,14,16,17,18,19,20]. In our study, we performed MRI only in those cases when implant rupture could not be confirmed by US. This could be considered as one of the limitations of our study as the number of such cases was low (*n* = 26); however, the main aim of our study was diagnostic discrimination of US in the evaluation of implant integrity. Our findings showed that US had a sensitivity of 98.3% and PPV of 93.4%. In different studies, the reported sensitivity and PPV of US in diagnostics of implant rupture vary widely, i.e., from 50% to 94% and from 52.4% to 90%, respectively [3,14,16,17,18,19,20]. Telegrafo and Moschetta analyzed the sensitivity of US to detect intracapsular and extracapsular implant ruptures separately and together. The sensitivity of US for extracapsular implant rupture was 100%, while for intracapsular rupture it was only 63% [18]. In our study, false-positive findings were documented in 4.2% of the cases, and despite the number of these cases was low, there was no need for operative treatment in all these cases. In order to avoid unnecessary surgery, inconclusive cases (when there are doubts if an implant is intact or ruptured on US) could be followed up by US every 3–6 months or MRI could be performed, despite the PPV of MRI varies from 78% to 98% [3,14,16,17,18,19,20]. Diagnostic discrimination of US and MRI of the current and other studies is shown in Table 5.

Such different results of US examination could be influenced by radiologist’s work experience. In some previous studies, it was indicated that US was performed by one or two experienced, highly specialized, and full-time breast diagnostics-dedicated radiologists [14,16,18,19]; however, other publications did not provide any information about the operator’s work experience [3,17,20]. The study by Hold et al. (radiologists’ experience was not reported) demonstrated the lowest sensitivity of US being 50% [17]. Benedetto et al. and Rietjens et al. (one experienced radiologist) reported the sensitivity higher by ~20–30% than in the above-mentioned studies—77% and 68.8%, respectively [14,16]. In the study by Bogetti et al., where images were interpreted by one experienced radiologist, the sensitivity was 90%, whereas Satti et al. (radiologist’s experience not specified) reported a higher sensitivity being 94% [19,20]. In the current study, the sensitivity was very high (98.3%), being the highest one in comparison with other studies and close to the sensitivity of MRI in the studies by Goldammer et al. and Satti et al. (98% and 99%, respectively) [3,20]. In our study, US examinations were performed by the experienced, highly specialized radiologist full-time dedicated to breast radiology and in part, to augmented breast examinations.

In our study, the specificity of US examination was high (89.2%), while in other studies, it varied from 55% to 90% [3,14,16,17,18,19,20].

The accurate diagnostics of the intact implant is also very important. The NPV of US in other studies varied from 58% to 93% [3,14,16,17,18,19,20]. In our study, the NPV of US was very high (97.1%) and close to the NPV of MRI in some previous studies [3,14,20]. False-negative results occurred in 1.1% of the operated cases and only in 0.7% of all cases. These data suggest that US is reliable and instead of MRI could become the “gold” standard in implant integrity assessment.

We analyzed the status of axillary lymph nodes because silicone-containing lymph nodes indicate a possible extracapsular implant rupture. In 7.9% (*n* = 12) of the 152 cases, silicones in axillary lymph nodes and a ruptured implant were confirmed after surgery. However, in 1.3% (*n* = 2) of the cases, silicone was found in axillary lymph nodes, but the implant was intact. According to the literature, a small amount of silicone gel can migrate from an implant even if it is not damaged, and this phenomenon is called silicone gel bleed [11,25]. Our patients with silicone-containing lymph nodes and an intact implant, however, had had previous implant damages and reoperations. Therefore, silicon deposits in lymph nodes could be related to previous implant ruptures.

In our study, MRI was performed only in 17.1% (*n* = 26) of all cases, and there were no false-negative results. Therefore, the NPV of MRI was 100% and was similar to the NPV (99%) reported in the study by Goldammer et al. [3]. The PPV of MRI in our study was 91.7% (*n* = 12) (one implant was flipped over, but not ruptured), while in other studies, it varied from 78% to 98% [3,14,16,17,18,19,20].

Our additional objective was to identify the main signs of an intact and a ruptured breast implant. On US, in the intact implant group, the shell was even and continuous as well as the content was homogeneous statistically more often than in the ruptured implant group (98.9% and 96.8% vs. 1.7% and 17.2%, *p* < 0.001). If both these signs of the ruptured implant were evident on US, MRI was not performed. In our study, of the 61 ruptured implants diagnosed by US, 48 (78.7%) had ≥2 signs, and the rupture was confirmed during surgery. When only one sonographic sign of a ruptured implant was found (uneven shell or inhomogeneous content), MRI was performed to verify implant integrity. The only sign by US and MRI (abnormal implant shell) was found in 10 cases (nine ruptured implants were confirmed during surgery). One implant with the signs of an abnormal shell was found to be not ruptured, but only flipped over; therefore, it is very important to know the signs of a flipped breast implant to avoid false-positive findings and unnecessary surgery at the same time since usually plastic surgeons can manually manipulate flipped implants into their correct position. The inhomogeneous content alone by US and MRI was found in three cases (surgically all these implants were intact). Silicone implants may also contain impurities, or the silicone gel may begin to aggregate or solidify over time, and these features on US examination may create spurious echoes within the implant and thus mimic implant rupture [22]. Based on these findings, we can conclude that the most important US sign of a ruptured implant is an abnormal implant shell, and in the presence of only inhomogeneous content, women could be followed up with US over time without any urgency of surgery.

Usually, breast implants tend to rupture 10 to 15 years from primary implantation [26]. Hölmich et al. calculated that the overall rupture incidence rate for definite ruptures was 5.3 ruptures/100 implants per year (95% CI, 4.0–7.0) after primary surgery, and the rupture rate increased significantly with an increasing implant age [27]. Oliveira et al. reported that the risk of the presence of alterations on MRI examination increased 1.07 times with every year (OR = 1.07, *p* = 0.036) [28]. In our study, the median time of rupture after implantation was 8 years, and with every year after implantation, the possibility of breast implant rupture increased 1.101 times. Furthermore, the ruptured implant was detected at least after one year following implantation surgery; therefore, based on these findings, we recommend performing routine screening by US annually, while MRI is advocated when US findings on implant integrity are inconclusive. Rietjens et al. also suggest performing US every year, while MRI, every 5 years [14]. Hold et al. do not advocate the use of MRI if US shows positive results of implant rupture [17], and Bogetti et al. recommend relying only on US in case of extracapsular rupture and avoiding MRI [19].

Until 2020, the FDA recommendations suggested that women with silicone gel-filled breast implants should undergo MRI screening 3 years after implant surgery and every 2 years after in order to diagnose asymptomatic ruptures [29]. The newly updated FDA 2020 recommendations suggest that asymptomatic patients should receive MRI or US 5 to 6 years after surgery and later on, every 2 to 3 years [15]. Considering the high cost, longer duration, and availability issues of MRI, US could be used for more frequent routine screening, taken into consideration that implant ruptures are usually asymptomatic and can occur earlier than expected.

Our study had some limitations. First, they include its small sample size and retrospective design. Second, the study population was from a single institution, and US evaluation was performed by one experienced and full-time breast diagnostics dedicated radiologist. In the future, it would be useful to include more institutions as well as radiologists with different work experience and compare the obtained results. Finally, we did not have information on the implant manufacturer in all cases in order to evaluate which implants ruptured more frequently.

## 5. Conclusions

Our results show that US is a very reliable alternative in evaluating breast implant integrity and could be the investigation of choice for implant rupture, while MRI could be advocated only in inconclusive cases. Uneven implant shell was found to be the most important US sign of breast implant rupture. Based on the findings, we recommend performing a US examination after breast augmentation surgery with silicone gel-filled implants annually.

## Figures and Tables

**Figure 1 medicina-57-00440-f001:**
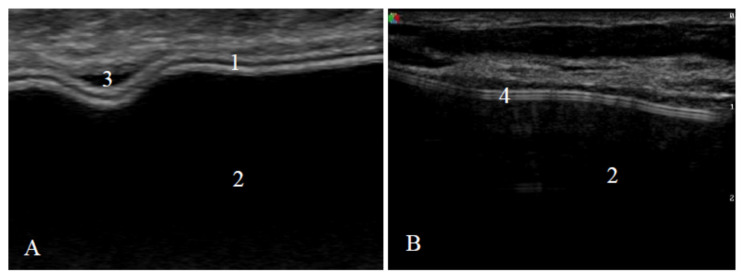
Signs of an intact implant on ultrasound: (**A**) a textured-surface implant (manufacturer Mentor) and (**B**) a smooth-surface implant (manufacturer Motiva); 1—shell (two parallel white lines); 2—homogeneous content; 3—implant shell wrinkle with small amount of peri-implant fluid; 4—shell (three parallel white lines).

**Figure 2 medicina-57-00440-f002:**
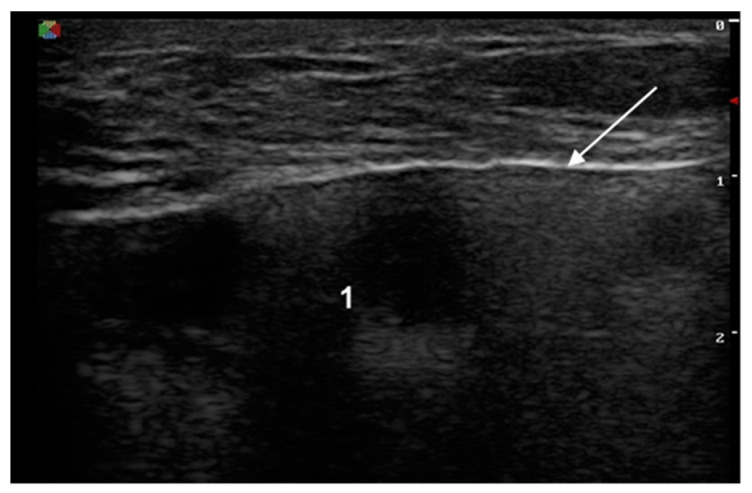
Signs of a ruptured implant on ultrasound: white arrow—abnormal shell (one white line is visible); 1—inhomogeneous content.

**Figure 3 medicina-57-00440-f003:**
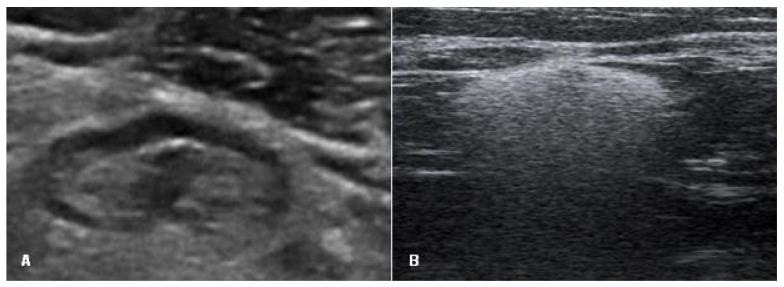
Signs of an axillary lymph node on ultrasound: (**A**) lymph node with normal appearance (oval shape, smooth cortex, unchanged, clearly visible fat gate) and (**B**) lymph node with silicone (a sharp upper border and sides and the loss of the lower border due to the “snowstorm” artifact).

**Figure 4 medicina-57-00440-f004:**
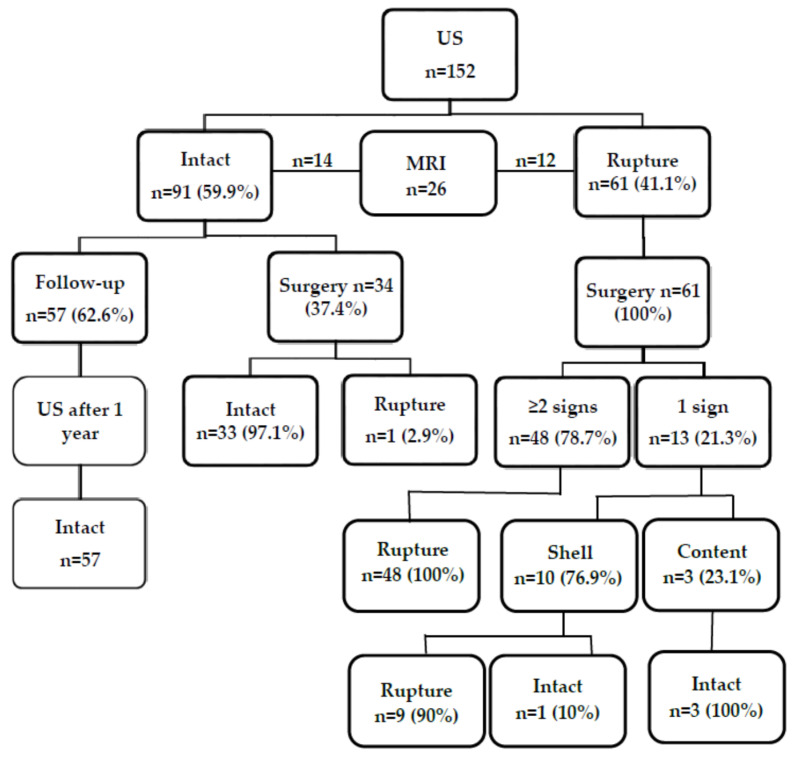
Flowchart showing the findings on ultrasound, magnetic resonance imaging, and at surgery.

**Figure 5 medicina-57-00440-f005:**
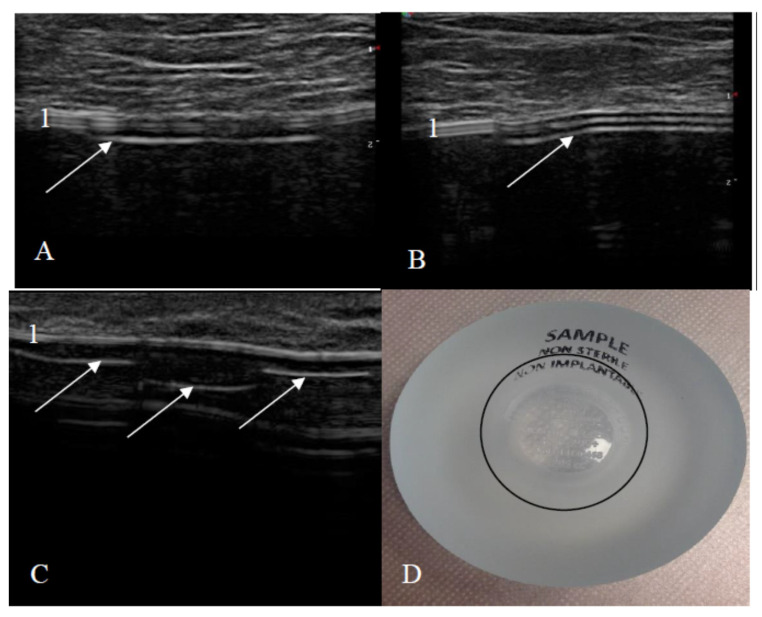
Breast implant displacement: signs of the flipped implant on ultrasound. (**A**,**B**) Images of the bottom of the same implant (manufacturer Motiva): (**A**) beginning of the implant bottom and (**B**) middle of the implant bottom; (**C**) only the bottom of the other implant (1—an intact shell, white arrows—the implant bottom looks like an abnormal shell: one additional line is seen only in the area of the implant bottom); and (**D**) natural image of the implant (a black circle indicates the implant bottom).

**Figure 6 medicina-57-00440-f006:**
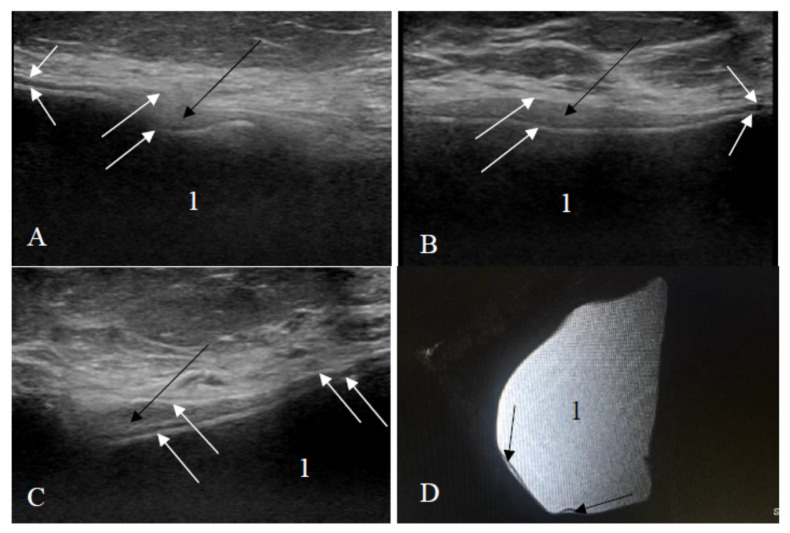
Ruptured implant (only one rupture sign visible): (**A**–**C**) ultrasound scans; (**D**) sagittal silicone-exited MRI sequence; white arrows—an uneven implant shell; black arrows—a small amount of silicone is outside the shell of the implant; 1—homogeneous content.

**Table 1 medicina-57-00440-t001:** Time since primary breast implant surgery.

Variable	Intact Implant	Ruptured Implant	*p* Value *	OR (95% CI)
Time since primary implant surgery, median (range; mean), years	4 (1–22; 5.5)	8 (1–22; 8.3)	0.001	1.101 (1.03–1.177)

* Mann—Whitney test. OR—odds ratio; CI—confidence intervals.

**Table 2 medicina-57-00440-t002:** Diagnostic value of US in all patients and operated cases.

	Acc % (95% CI)	Se % (95% CI)	Sp % (95% CI)	PPV % (95% CI)	NPV % (95% CI)	FP % (95% CI)	FN % (95% CI)
All cases (*n* = 152)	96.7 (93.1; 100.3)	98.3 (94; 102.5)	95.7 (90.5; 100.9)	93.4 (85.5; 101.4)	98.9 (96.2; 101.6)	2.6 (−0.6; 5.9)	0.7 (−1; 2.3)
Operated cases (*n* = 95)	94.7 (89; 100.5)	98.3 (86; 99.9)	89.2 (68.8; 97.3)	93.4 (85.5; 101.4)	97.1 (89.8; 104.3)	4.2 (−0.9; 9.4)	1.1 (−1.6; 3.7)

US, ultrasound; Acc, accuracy; Se, sensitivity; Sp, specificity; PPV, positive predictive value; NPV, negative predictive value; FP, false positive; FN, false negative; CI, confidence interval.

**Table 3 medicina-57-00440-t003:** Diagnostic signs of intact and ruptured implants on ultrasound.

Sign		Implant, n (%)		*p * Value *	OR (95% CI)
		Intact	Ruptured		
Shell ^a^	Normal	93 (98.9)	1 (1.7)	<0.001	5301.0 (325.156–86422.01)
	Abnormal	1 (1.1)	57 (98.3)		
Content ^b^	Homogeneous	91 (96.8)	10 (17.2)	<0.001	145.6 (38.249–554.243)
	Inhomogeneous	3 (3.2)	48 (82.8)		
Axillary lymph nodes ^c^	Normal	92 (97.9)	46 (79.3)	<0.001	12.0 (2.577–55.875)
	Abnormal	2 (2.1)	12 (20.7)		

* Fisher’s exact test; ^a^ normal implant shell as a reference category; ^b^ homogeneous implant content as a reference category; ^c^ normal axillary lymph nodes as a reference category. OR, odds ratio; CI, confidence interval.

**Table 4 medicina-57-00440-t004:** Distribution of intact and ruptured implants by the number of signs of an intact and a ruptured implant on ultrasound.

Implant	Number of Signs on US			
	0	1	2	3
Intact ^a^	0	0	6 (6.4)	88 (93.6)
Ruptured ^b^	1 (1.7)	9 (15.5)	36 (62.1)	12 (20.7)

Values are number (percentage). ^a^ The signs of an intact implant are even implant shell, homogeneous content, structural axillary lymph nodes; ^b^ the signs of a ruptured implant are uneven implant shell, inhomogeneous content, silicone-containing axillary lymph nodes.

**Table 5 medicina-57-00440-t005:** Comparison of diagnostic discrimination of ultrasound and magnetic resonance imaging.

Author	S	R	N	Method	Se %	Sp %	PPV %	NPV %
Di Benedetto et al., 2008 [16]	r	1	82	US	77	69	84	58
				MRI	93	73	88	82
Hold et al., 2012 [17]	r	–	60	US	50	90	86	–
				MRI	83	90	91	–
Rietjens et al., 2014 [14]	p	1	102	US	68.8	73.3	52.4	84.6
				MRI	82.9	97.8	93.5	93.6
Telegrafo and Moschetta, 2015 [18]	p	2	300	US	79	63	65	77
				US extr	100	100	100	100
				US intr	63	63	45	77
Bogetti et al., 2018 [19]	r	1	51	US	90	80	88	84
				MRI	87	85	90	81
Satti et al., 2020 [20]	r	–	60	US	94	55	90	67
				MRI	98	91	98	91
Goldammer et al., 2020 [3]	r	–	295	US	84	90	78	93
			160	MRI	99	78	78	99
Our study, 2021	r	1	152	US	98.3	89.2	93.4	97.1
			26	MRI	100	93.3	91.7	100

S, study; r, retrospective; p, prospective; R, radiologist; –, no information; 1, 2, one or two radiologists with experience in breast imaging; US, ultrasound; MRI, magnetic resonance imaging; Se, sensitivity; Sp, specificity; PPV, positive prognostic value; NPV, negative prognostic value.

## Data Availability

The data presented in this study are available on request from the corresponding author. The data are not publicly available due to ethical restrictions and data protection policies.

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
