# Peer review of "The Value of Ultrasound in the Evaluation of the Integrity of Silicone Breast Implants"

_medicina, 2021, doi:10.3390/medicina57050440_

Round 1
Reviewer 1 Report
Comments to Author:
I read with interest this article “The value of ultrasound in the evaluation of the integrity of breast silicone implants”.
The manuscript describes the use of US in the evaluation of breast implant rupture. The main object of the study is to determine if US is reliable in the evaluation of breast implant integrity.
Considerations:
- No ethical approvals have been stated. Which IRB approved the study? Date? Reference number? Was this trial registered (e.g., clinicaltrials.gov) ?
- There is no mention of breast implant type (smooth/textured)? Did you note any differences between implants brand?
- Did you note any differences between subglandular/subfascial/submuscular implant signs of rupture?
- You should discuss the limitations of the study (number of patient, restrospective design of the study)
- as reported in literature (Zingaretti N et al. Determinants of surgeon choice in cases of suspected implant rupture following mastectomy or aesthetic breast surgery: clinical implications. Medicine. 2020; 99: e21134), from a medico-legal point of view, when do you think MRI is mandatory? only in inconclusive cases?
-Finally, you need to be very clear about what this adds to the existing literature and clearly detail learning points.
I believe if you’ll add these informations this report can be of interest for Journal readers.
Author Response
Response to Reviewer 1 Comments
First at all, we are very grateful for all reviewer’s comments that helped to improve our manuscript. All the changes done in the manuscript are highlighted in red.
Point 1: No ethical approvals have been stated. Which IRB approved the study? Date? Reference number? Was this trial registered (e.g., clinicaltrials.gov)?
Response 1: The information regarding ethical approval was stated at the beginning of Materials and methods. In addition, we added this information at the end of the manuscript. This study was not registered at clinicaltrials.gov.
Point 2: There is no mention of breast implant type (smooth/textured)? Did you note any differences between implants brand?
Response 2: Unfortunately, this can be considered as one of the limitations of our study. As we knew the manufacturer of implants only in 51 cases (33.6%), no analysis on the incidence of implant rupture by the manufacturer could be done. Women who were enrolled in our study underwent breast reconstruction surgery in other plastic and reconstructive surgery clinics as well, and simply on admission to our clinic could not provide any information on the implant manufacturer. Also, no data on the breast implant type (smooth/textured) were collected.
Point 3: Did you note any differences between subglandular/subfascial/submuscular implant signs of rupture?
Response 3: The information on implant localization was not collected. However, as the main signs of implant rupture are changes in the implant shell and content, there should be no any difference.
Point 4: You should discuss the limitations of the study (number of patient, retrospective design of the study)
Response 4: Following the reviewer’s comment, these limitations were included in the discussion as well.
Point 5: As reported in literature (Zingaretti N et al. Determinants of surgeon choice in cases of suspected implant rupture following mastectomy or aesthetic breast surgery: clinical implications. Medicine. 2020; 99: e21134), from a medico-legal point of view, when do you think MRI is mandatory? only in inconclusive cases?
Response 5: Zingaretti et al. recommend performing MRI in all cases when implant rupture is identified. We think that if implant rupture can be clearly identified on US (US image shows an abnormal implant shell and inhomogeneous content), no MRI should be performed. However, radiologist’s work experience should not be ignored. We recommend performing MRI only in inconclusive cases, i.e. when only one sign of implant rupture is found.
Point 6: Finally, you need to be very clear about what this adds to the existing literature and clearly detail learning points.
Response 6: Following the reviewer’s suggestion, the text at the beginning of the discussion was amended and learning points have been added.
“This study had a dual aim: 1) to determine the diagnostic value of US in the evaluation of implant integrity and 2) to identify which US signs of an intact and a ruptured implant are most important. Our study showed that US is very accurate in the evaluation of implant integrity (diagnostic accuracy of 94.7%, sensitivity of 98.3%, NPV of 97.1%) and it has to be a first-choice imaging modality. We recommend performing US every year after breast augmentation surgery as an implant can rupture earlier than it is expected. If ≥ 2 signs of a ruptured implant are seen on US, we can rely on US examination by 100%. If only one sign of a ruptured implant is found, it is advocated to perform MRI as well. The most important sign of a ruptured implant on US is an abnormal implant shell as the implant content can become inhomogeneous without any implant damage.”

Reviewer 2 Report
MAJOR
1 - This is a study where the authors claim to validate that ultrasound is highly predictive of ruptured implants, comparable to MRI which is considered the gold standard. To prove their point, they select cases with ruptured implants and compared with cases without ruptured implants and conclude that US is highly sensitive, highly specific, as long as an experienced radiologist identifies the signs of rupture. However, my main concern is that the authors are highly skewed in their case selection, suggesting that the only 2 diagnostic options when evaluating US on patients with breast implants are ruptured or not ruptured. In reality, there is a wider spectrum of disorders that occur in patients with breast implants, such as contracture, effusion for unknown reasons, hemorrhage, or lymphoma. Therefore, to consider ultrasound as a valid and useful approach to diagnosis, a wider range of disorders need to be included in the evaluation and validation of US in patients with implants.
2 - An important consideration for the authors to ponder in this study is the validation of ruptured implants. How was ruptured defined? What were the criteria for a physician to determine the implant is ruptured? Was there a pathologic evaluation of the implants and changes in the surrounding capsules that reflected rupture? Was there foreign body giant cell reaction or Silicone particles in the capsules?
3 - Abstract: Not clear: “Of them (are the authors referring to the 13 cases with only one sign of ruptured implant?, or to the total # of cases with US evidence of rupture?) an abnormal implant shell was documented more often … (n=9, 90% vs n=1, 10% … (where are these numbers coming from? The authors stated that 61 implants were ruptured by Ultrasound.
MINOR
1-Introduction: “free silicone travels to distant regions, i.e., chest, upper extremities…” chest is not far from breast… please rewrite sentence
2- “gold” standard instead of “golden”
3- Silicone implants: Are not all implants of silicone? The so-called implant rupture should be referred to Silicone-filled as opposed to “saline filled” implants.
Author Response
Response to Reviewer 2 Comments
First at all, we are very grateful for all reviewer’s comments that helped to improve our manuscript. All the changes done in the manuscript and our answers are highlighted in red.
MAJOR
Point 1: This is a study where the authors claim to validate that ultrasound is highly predictive of ruptured implants, comparable to MRI which is considered the gold standard. To prove their point, they select cases with ruptured implants and compared with cases without ruptured implants and conclude that US is highly sensitive, highly specific, as long as an experienced radiologist identifies the signs of rupture. However, my main concern is that the authors are highly skewed in their case selection, suggesting that the only 2 diagnostic options when evaluating US on patients with breast implants are ruptured or not ruptured. In reality, there is a wider spectrum of disorders that occur in patients with breast implants, such as contracture, effusion for unknown reasons, hemorrhage, or lymphoma. Therefore, to consider ultrasound as a valid and useful approach to diagnosis, a wider range of disorders need to be included in the evaluation and validation of US in patients with implants.
Response 1: Yes, we totally agree that there are more implant complications; however, most commonly they have clinical manifestation. Therefore, as in majority of cases, implant ruptures are asymptomatic, in this study we focused only on implant ruptures and diagnostic value of ultrasound in identifying implant ruptures. If implant rupture is not timely diagnosed, free silicone can migrate that leads to the formation silicone granulomas in different locations. Therefore, we consider this complication as one of the most dangerous.
Coming back to other implant complications, we would like to outline the typical clinical manifestations of these complications:
- a) Implant contracture – the signs of capsular contracture of degree 1 and 2 are only clinical: pain in the breast and the breast become firm. Usually patients with these symptoms refer to a plastic surgeon gets help needed. On ultrasound, only contractures of degree 3 and 4 can be seen.
- b) Effusion and hemorrhage also have clinical manifestation. Fluid accumulation around the implant leads to the increased breast volume and patients refer to plastic surgeons.
- c) Lymphoma (did you have anaplastic large-cell lymphoma in your mind?) – often the clinical manifestations of this disease are late seroma and increased breast volume. However, no single case of this disease has been documented in Lithuania.
As our aim was to determine the diagnostic discrimination of ultrasound only in implant ruptures, no other disorders were considered in our study.
Point 2: An important consideration for the authors to ponder in this study is the validation of ruptured implants. How was ruptured defined? What were the criteria for a physician to determine the implant is ruptured? Was there a pathologic evaluation of the implants and changes in the surrounding capsules that reflected rupture? Was there foreign body giant cell reaction or Silicone particles in the capsules?
Response 2: Once implant rupture was detected by US and/or MRI, it was followed by surgery. If intraoperatively the implant shell was undamaged and there was no free silicone outside the implant shell, the implant was defined as intact, and if there was a shell defect and silicone was present outside the implant shell, the implant was considered as ruptured. Once a woman arrives to the clinic, the attending surgeon refers her to a radiologist. The radiologist based on ultrasound findings decides if there is rupture or not. No data on siliconomas were collected.
Point 3: Abstract: Not clear: “Of them (are the authors referring to the 13 cases with only one sign of ruptured implant?, or to the total # of cases with US evidence of rupture?) an abnormal implant shell was documented more often … (n=9, 90% vs n=1, 10% … (where are these numbers coming from? The authors stated that 61 implants were ruptured by Ultrasound.
Response 3: Yes, we are referring to the 13 cases with only one sign of ruptured implant, i.e. only an abnormal implant shell or inhomogeneous implant content. If only inhomogeneous implant content was found (n=3), in all cases, the implants were found to be intact. If an abnormal implant shell was found (n=10), one implant was intact (10%) and 9 (90%) implants were ruptured. In total, this makes 13 cases.
For clarity we amended the text in the manuscript as well.
MINOR
Point 1: 1-Introduction: “free silicone travels to distant regions, i.e., chest, upper extremities…” chest is not far from breast… please rewrite sentence
Response 1: It was changed as follows: “Occasionally, free silicone travels to distant regions (arm/forearm, thoracic cavity, abdominal wall, legs, back) [9, 10].”
Point 2: “gold” standard instead of “golden”
Response 2: Thank you for noting this mistake. The amendment is done.
Point 3: Silicone implants: Are not all implants of silicone? The so-called implant rupture should be referred to Silicone-filled as opposed to “saline filled” implants.
Response 2: We are not sure if we correctly understood the reviewer’s question. Is the reviewer asking to change the phrase “implant rupture” to “silicone gel-filled implant rupture” in the entire manuscript? If yes, we do not know if this should be done as in the title of our manuscript we refer to “silicone breast implants” and this implies that no other type of implants was analyzed in our study. However, for clarity, we added “silicone” at the beginning of the abstract and “silicone gel-filled” in the conclusions.

Reviewer 3 Report
The authors have determined ultrasound as an important tool to suspect rupture in a breast implant, in cases where an MRI is not feasible or not cost-effective. Many such studies have already been published in different clinical and research journals. For example:
Bengtson, Bradley P., and Felmont F. Eaves III. "High-resolution ultrasound in the detection of silicone gel breast implant shell failure: background, in vitro studies, and early clinical results." Aesthetic surgery journal 32.2 (2012): 157-174.
Ikeda, Debra M., et al. "Silicone breast implant rupture: pitfalls of magnetic resonance imaging and relative efficacies of magnetic resonance, mammography, and ultrasound." Plastic and reconstructive surgery 104.7 (1999): 2054-2062.
Rietjens, Mario, et al. "Appropriate use of magnetic resonance imaging and ultrasound to detect early silicone gel breast implant rupture in postmastectomy reconstruction." Plastic and reconstructive surgery 134.1 (2014): 13e-20e.
Bengtson, Bradley P., and Felmont F. Eaves III. "High-resolution ultrasound in the detection of silicone gel breast implant shell failure: background, in vitro studies, and early clinical results." Aesthetic surgery journal 32.2 (2012): 157-174.
Evans, Andrew, et al. "Breast ultrasound: recommendations for information to women and referring physicians by the European Society of Breast Imaging." Insights into imaging 9.4 (2018): 449-461.
Pizzolon, María Flavia, Marcela Uchida, and Eugenio Soto. "Breast implants late collections, ultrasound approach." European Congress of Radiology-ECR 2019, 2019.
Thus this study is deemed not novel and just a repetition of the already published studies.
Author Response
Response to Reviewer 3 Comments
We are very grateful for the comments by the reviewer and his/her time as well as efforts given to review our manuscript. Please find our rebuttal below.
Point 1: The authors have determined ultrasound as an important tool to suspect rupture in a breast implant, in cases where an MRI is not feasible or not cost-effective. Many such studies have already been published in different clinical and research journals. For example:
1) Bengtson, Bradley P., and Felmont F. Eaves III. "High-resolution ultrasound in the detection of silicone gel breast implant shell failure: background, in vitro studies, and early clinical results." Aesthetic surgery journal 32.2 (2012): 157-174.
2) Ikeda, Debra M., et al. "Silicone breast implant rupture: pitfalls of magnetic resonance imaging and relative efficacies of magnetic resonance, mammography, and ultrasound." Plastic and reconstructive surgery 104.7 (1999): 2054-2062.
3) Rietjens, Mario, et al. "Appropriate use of magnetic resonance imaging and ultrasound to detect early silicone gel breast implant rupture in postmastectomy reconstruction." Plastic and reconstructive surgery 134.1 (2014): 13e-20e.
4) Bengtson, Bradley P., and Felmont F. Eaves III. "High-resolution ultrasound in the detection of silicone gel breast implant shell failure: background, in vitro studies, and early clinical results." Aesthetic surgery journal 32.2 (2012): 157-174.
5) Evans, Andrew, et al. "Breast ultrasound: recommendations for information to women and referring physicians by the European Society of Breast Imaging." Insights into imaging 9.4 (2018): 449-461.
6) Pizzolon, María Flavia, Marcela Uchida, and Eugenio Soto. "Breast implants late collections, ultrasound approach." European Congress of Radiology-ECR 2019, 2019.
Thus this study is deemed not novel and just a repetition of the already published studies.
Response 1: We totally agree that there are some studies published on this topic. However, the results reported in these studies vary considerably; therefore, there is no single answer and this prompts new studies. This could be confirmed by studies performed in recent years demonstrating the relevance of such research. Other thing worth mentioning is that there are no studies investigating the value of particular signs of implant rupture on ultrasound in order to find the most important ones. Finally, there are no data on implant rupture in Lithuania and this is the first study analyzing the diagnostic value of ultrasound in implant rupture in the Lithuanian population.
From the list of the references provided by the reviewer, one could see that references no. 1 and 4 is the same study (Bengtson, Bradley P., and Felmont F. Eaves III. "High-resolution ultrasound in the detection of silicone gel breast implant shell failure: background, in vitro studies, and early clinical results." Aesthetic surgery journal 32.2 (2012): 157-174); reference no. 5 (Evans, Andrew, et al. "Breast ultrasound: recommendations for information to women and referring physicians by the European Society of Breast Imaging." Insights into imaging 9.4 (2018): 449-461.) is the recommendations on breast ultrasound by the European Society of Breast Imaging; reference no. 6 (Pizzolon, María Flavia, Marcela Uchida, and Eugenio Soto. "Breast implants late collections, ultrasound approach." European Congress of Radiology-ECR 2019, 2019) is a poster dealing with the prevalence of complications such capsular contracture and implant failure; late seromas, spontaneous hematoma and implant-associated anaplastic large cell lymphoma. In the last one, the authors did not aim at determining the diagnostic accuracy of any imaging modality; therefore, this study cannot be considered as applicable in our case as well. After excluding these literature sources, it leaves us with 3 publications on the same topic.
Summarizing, we believe that our study contributes to the existing knowledge and adds some new data.

Reviewer 4 Report
Dear authors,
Your manuscript is interesting as more and more women decide to have breast augmentation surgery. Even if they are not aware of the possible complications, such as BIA-ALCL, all patients should do an annually check up.
It is true that MRI is expensive and difficult to perform each year, so the US examination remains the perfect option.
The study was conducted just on women with breast augmentation. What about the patients with breast reconstruction based on implants?
Overall the article is well written, although the references should be from the last 10 years.
Author Response
Response to Reviewer 4 Comments
First at all, we are very grateful for the reviewer’s comments that helped to improve our manuscript. All the changes done in the manuscript are highlighted in red.
Point 1: The study was conducted just on women with breast augmentation. What about the patients with breast reconstruction based on implants?
Response 1: In Lithuania, breast reconstruction surgeries due to breast cancer in a public health care sector are covered by patient funds, and patients have to pay only for an implant if they choose implant reconstruction rather than autologous reconstruction. In a private health care sector, neither surgery nor an implant is covered. Tautrimas Aštrauskas Clinic is a private clinic; therefore, only a limited number of patients refer to this clinic for reconstruction surgery. In order to make the study population more homogeneous, we recruited only patients who underwent bilateral breast reconstruction surgery due to cosmetic reasons.
Point 2: Overall the article is well written, although the references should be from the last 10 years.
Response 2: After removing one reference from the list based on the suggestion by other reviewer, there are only 4 references older than 10 years. This accounts for a small percentage of the total number.

Reviewer 5 Report
In the present draft the authors present a retrospective study to evaluate the diagnostic value of ultrasound (US) in evaluating silicone breast implant integrity and to identify which US indications of intact/ruptured implants are most important. The authors show US to be a reliable alternative in monitoring implant rupture over the more expensive MRI, which the authors suggest only be employed in inconclusive cases. Moreover, an uneven implant shell was found to be the most important US indication of rupture.
Overall, data are interesting and are associated with a certain degree of novelty, and the topic discussed and the findings are compatible with the area of interest of the journal.
Please find comments below that need addressing:
- Line 61 – please define US on FIRST use. You do do this in line 62, but it should be done here then use abbreviated form hereafter
- M&M and Results – please add subheadings for both sections to improve clarity and flow
- M&M:
- Please add the data acquisition features used for US and MRI. Furthermore, what optimisation, if any, was used with the US images, given the differing image qualities eg Fig 1 vs Fig 2.
- Fig 1A and B – it would be better to show the whole US image or more of the implant structure as you did for Figs 2 and 5.
- Fig 1 legend – please add more detail either here or in the main text regarding the significance of parallel white lines
- Line 121 – how many US check-ups were needed to confirm rupture for the 48 patients?
- Results
- Figure 4 should be referred to in the opening lines of the results section
- Figure 5 – is this a textured implant? Also in legend, please clarify “implant bottom looks like uneven shell”, as this reviewer sees no difference between this image and those in Fig 1
- Figure 6 legend, “white arrows – abnormal implant shell, a small amount of silicone…”. The same could also be said of Fig 1 intact implants, please clarify. Also where are the black arrows showing silicone outside the implant shell (line 212)?
- Lines 218-222 – please rephrase
- Table 5/Line 248 – Please remove comparisons to Everson et al. study as it was over 20 years ago
- Few typographical mistakes must be fixed (eg line 217 “sings”). Please check the entire manuscript
Author Response
Response to Reviewer 5 Comments
First at all, we are very grateful for all reviewer’s comments that helped to improve our manuscript. All the changes done in the manuscript are highlighted in red.
Point 1: Line 61 – please define US on FIRST use. You do do this in line 62, but it should be done here then use abbreviated form hereafter

Response 1: Now US is defined on the first use.
M&M
Point 2: Please add the data acquisition features used for US and MRI.
Response 2: The information on the data acquisition features used for US and MRI was updated as follows:
“US examinations were performed with a SonoScape S6/S6Pro/S6BW US unit using a 12-MHz linear transducer at two-dimensional (B2) mode. MRI was performed using an MRI scanner (Siemens MAGNETOM® Avanto 1.5 T) with a dedicated breast coil. The following MRI sequences for the examination of implant morphology and integrity were acquired: 1) the high-resolution T2-weighted axial sequences; 2) water-suppressed fast spin echo T2-weighted axial sequences; and 3) the axial inversion recovery (IR) sequence with water saturation (”silicone only”) sequences. Sagittal sequences were acquired on the side of the implant being investigated. No oral or intravenous contrast was given.”
Point 3: Furthermore, what optimisation, if any, was used with the US images, given the differing image qualities eg Fig 1 vs Fig 2.
Response 3: Based on the reviewer’s comment, we have changed the images for better quality.
Point 4: Fig 1A and B – it would be better to show the whole US image or more of the implant structure as you did for Figs 2 and 5.
Response 4: Based on the reviewer’s comment, we have changed Figures 1A and B.
Point 5: Fig 1 legend – please add more detail either here or in the main text regarding the significance of parallel white lines
Response 5: We updated the description of Figure 1 and the figure itself. Regarding the significance of parallel while lines we must say that the shell of intact implants produced by the majority of manufacturers looks like two parallel white lines on US, while the shell of intact implants by manufacturer Motiva appears on US as three parallel white lines. If a US image reveals the expansion and loss of parallel running lines or so-called “stepladder” sign, then implant rupture is suspected.
Point 6: Line 121 – how many US check-ups were needed to confirm rupture for the 48 patients?
Response 6: Unfortunately, we do not have such data. Women who have breast implants do not tend to have regular check-ups. In all these 48 patients both signs of implant rupture (abnormal shell and inhomogeneous implant content) were found on US; therefore, the conclusion of US examination was implant rupture and US was not repeated. The patients underwent surgery and implant rupture was confirmed during surgery.
Results
Point 1: Figure 4 should be referred to in the opening lines of the results section
Response 1: Following the reviewer’s suggestion, now Figure 4 is referred in the opening lines of Results.
Point 2: Figure 5 – is this a textured implant? Also in legend, please clarify “implant bottom looks like uneven shell”, as this reviewer sees no difference between this image and those in Fig 1
Response 2: Following the reviewer’s comment, the figure with the descriptions was changed for the clarity.
Point 3: Figure 6 legend, “white arrows – abnormal implant shell, a small amount of silicone…”. The same could also be said of Fig 1 intact implants, please clarify. Also where are the black arrows showing silicone outside the implant shell (line 212)?
Response 3: Following the reviewer’s comment, the figure with the descriptions was changed for clarity. During the conversion of Word to pdf, it might be that the black arrow was lost. Now we made sure that all the arrows and other signs are present.
Point 3: Lines 218-222 – please rephrase
Response 3: We rephrased the indicated lines in the discussion as follows:
“This study had a dual aim: 1) to determine the diagnostic value of US in the evaluation of implant integrity and 2) to identify which US signs of an intact and a ruptured implant are most important. Our study showed that US is very accurate in the evaluation of implant integrity (diagnostic accuracy of 94.7%, sensitivity of 98.3%, NPV of 97.1%) and it has to be a first-choice imaging modality. We recommend performing US every year after breast augmentation surgery as an implant can rupture earlier than it is expected. If ≥ 2 signs of a ruptured implant are seen on US, we can rely on US examination by 100%. If only one sign of a ruptured implant is found, it is advocated to perform MRI as well. The most important sign of a ruptured implant on US is an abnormal implant shell as the implant content can become inhomogeneous without any implant damage.”
Point 4: Table 5/Line 248 – Please remove comparisons to Everson et al. study as it was over 20 years ago
Response 4: The reference indicated was removed from the table as well as from the whole manuscript.
Point 5: Few typographical mistakes must be fixed (eg line 217 “sings”). Please check the entire manuscript
Response 5: The whole manuscript was revised for typographical mistakes.

Round 2
Reviewer 2 Report
Queries have been addressed.
Author Response
Thank you for your time and efforts in reviewing our manuscript.
Reviewer 3 Report
Most comments have been addressed by the authors
Author Response
Thank you for your time and efforts in reviewing our manuscript. As the reviewer has not pointed out what comments are left unaddressed, we really do not know how further to improve our manuscript.
Reviewer 5 Report
The following is yet to be addressed in the manuscript:
“Results
Point 1: Figure 4 should be referred to in the opening lines of the results section
Response 1: Following the reviewer’s suggestion, now Figure 4 is referred in the opening lines of Results “
Otherwise reviewers concerns have been addressed.
Author Response
Please accept our apologies for not properly addressing this issue. Now Figure 4 is cited in the first sentence of the results. The change is highlighted in green color.